# Spatial Pattern and Influencing Factors of Basic Education Resources in Rural Areas around Metropolises—A Case Study of Wuhan City's New Urban Districts

**Liang Jiang [1], Jie Chen [2,*], Ye Tian [3] and Jing Luo [4]**

1. College of Urban and Environmental Sciences, Xuchang University, Xuchang 461000, China
2. School of Geography and Tourism, Huizhou University, Huizhou 516007, China
3. Institute for Advanced Studies in Finance and Economics, Hubei University of Economics, Wuhan 430205, China
4. College of Urban and Environmental Sciences, Central China Normal University, Wuhan 430079, China
* Correspondence: chenjie@hzu.edu.cn; Tel.: +86-176-0718-8530

**Abstract:** Basic education resources are basic urban and rural social public security resources, and their spatial distribution is an important issue related to people's livelihoods and social justice. Taking Wuhan as a case study, this paper analyzed the spatial distribution characteristics of rural basic education resources based on the methods of the average nearest neighbor index, imbalance index, kernel density analysis and two-step floating catchment area and then used geographic detector analysis to detect its influencing factors. The following findings were obtained: (1) Rural kindergartens and elementary schools in Wuhan City's new urban districts showed a clustered distribution pattern, while secondary schools showed a uniform distribution trend. The spatial distribution of rural basic education resources is poorly balanced, with a tendency to cluster in Huangpi District, Xinzhou District and Caidian District; the overall spatial distribution density of rural basic education resources showed the distribution characteristics of "block-like clustering and multicenter development". (2) The spatial accessibility of kindergartens showed a spatial pattern of "large dispersion and small clustering", with multiple high-value clustering areas; and the accessibility of elementary and secondary schools showed a spatial pattern of high in the south and low in the north. (3) The population, economy and education development level are the main factors affecting the spatial distribution of rural basic education resources, while the influence of infrastructure construction is weak. The core influencing factors of the spatial distribution of each type of basic education resource are both consistent and different. According to the interaction factor detection, the spatial distribution of rural basic education resources in Wuhan City's new urban districts is the result of the combined effect of multiple factors.

**Keywords:** rural basic education; spatial pattern; influencing factors; geographic detector; Wuhan City's new urban districts

## 1. Introduction

Basic public services, including public services such as education, medical care, employment and pensions, are the minimum public services and facilities that meet the needs of all citizens [1]. The rapid development of the social economy, especially the knowledge economy [2], has made education increasingly important in social progress and development. Equality of access to quality education is particularly important for enhancing short- and long-term outcomes alike. Participating in education has far-reaching implications for fundamental areas such as labor market opportunities, intergenerational mobility and health [3–5]. By the end of 2020, there were 537,100 schools at all levels, and 289 million students enrolled in all academic levels in China [6]. Education, as the cornerstone of national revitalization and social progress, serves as a key to promoting the overall

quality of the population and the sustained improvement of the country's comprehensive strength [7]. According to the United Nations, it is projected that the proportion of the world's population living in urban areas will increase to 68% by 2050 from 55% in 2018, and China is expected to add 255 million urban dwellers [8]. However, the rational spatial allocation of educational resources has resulted in serious challenges for urban and rural planning and management. Although Chinese government started the Rural School Mapping Adjustment (RSMA) by withdrawing and merging most village-level schools that were considered to be inefficient and costly, promoting boarding schools for rural basic education, shifting primary schools from the village to the township and shifting secondary schools from the township to the county [9], this actually makes the size of county and town schools show a trend of expansion, and the basic education in rural areas shrinks, exacerbating the equality and efficiency issues caused by the expansion of school size in the context of rural school layout adjustment [10,11]. It should be emphasized that educational equality can help provide educational resources to every student [12]. With the rapid growth of the population in metropolitan areas, the demand for infrastructure greatly exceeds the service capacity of existing service facilities [13,14]. The shortage of educational facilities in the suburbs and rural areas often results in unfair education distribution.

Education is becoming a focus issue, and a good education system is important to eliminate social inequity and poverty [15,16]. Specifically, rural education is one of the important factors in regional development supported by the central government. At present, despite the universal, compulsory education policy in China's urban and rural areas, there are still disparities in the quantity and level of basic education resources among regions, and between urban and rural areas and schools, highlighting the contradiction between the people's growing need for quality education and the insufficient supply [17]. In the report of the 19th National Congress, the concepts of "striving to let every child enjoy a fair and quality education" and "promoting the integrated development of urban and rural compulsory education and attaching great importance to rural compulsory education" were put forward. These concepts reflect that education is an important element of rural socioeconomic development and coordinated urban–rural development and that educational equity and rural revitalization are top priorities [18]. More importantly, education is an important way to break the intergenerational transmission of poverty [19]. Because of the importance of promoting the integration of urban and rural compulsory education and equalizing urban and rural public education resources, it is necessary to study the spatial pattern of rural basic education resources and its influencing factors.

As early as the 1960s, researchers studied education resources in terms of structure and function, treating education as a productive element in the social structure and exploring the relationship between the distribution of educational resources and the macropolitical and economic structure. Blaug pointed out in his "approach to educational planning" that the allocation of educational resources in Western society is often tied to economic revitalization plans and the goal of economic development [20]. On this premise, researchers have explored the physical infrastructure, spatial layout, location selection, supply mode and efficiency of education resources [21–24]. Postmodernism has led to the evolution of human geography from spatial analysis to social theory [25], and along with this, the "sociality" of space has attracted increasing attention [26,27], focusing on the natural and human-social environmental mechanisms behind educational phenomena. Hones et al. proposed the "space-mechanism-effect" research framework of education resources. As this research framework organically combines the spatial pattern of education resources, evolutionary mechanisms and influence on the spatial pattern of social and cultural phenomena, it has become the main foundation of research on the spatial pattern of modern Western education resources [28]. The development of the 3S (RS, GIS, GPS) geographic information technology has brought scholars to explore spatial and temporal accessibility [29,30], equalization [31,32], spatial differentiation and equity evaluation of education resources [33,34] from different scales. Other researchers have also investigated how education resources

are antecedents that shape social structure and sociospatial heterogeneity, such as housing prices [35,36], urbanization processes [32] and individual achievement [37].

In China, scholars mainly focus on the evaluation index system, regional differences, spatial patterns and the optimal layout of education resources, and they are devoted to exploring the fairness of educational resource allocation, spatial equalization and planning rationality [38]. For example, by constructing a compulsory education resource index system from school conditions and teacher strength, Yu et al. found that the overall difference in the spatial pattern of compulsory education resources in Liaoning Province showed a gradually increasing trend [39]. Based on two spatial scales, interschool at the county level and interschool at the city level, Liu et al. found that elementary education resources in Dalian are highly balanced and contiguous among schools within the county as a whole but are poorly balanced within the city. Using methods such as hot spot analysis and kernel density analysis [40], Zhao et al. found that under the further integration of rural basic education resources, the quantitative scale of rural schools in Southwest China decreases while the spatial agglomeration increases significantly [41]. Peng et al. proposed a method for optimizing the layout of education resources relying on GIS spatial analysis and spatial operations research [42,43]. The in-depth discussion on the equity and balance of educational resources resulted in an increasing number of studies targeting the spatial accessibility of schools. Based on the minimum proximity method, the gravity model [44] and the two-step floating catchment area method [45], scholars have analyzed differences in the spatial distribution of urban and rural educational resources, explored their influencing factors [46–48], and proposed policy paths to address the imbalances [49,50].

Although existing studies on education are of great importance, there are some deficiencies. First, the current research on the layout and optimization of educational resources has produced fruitful results, but there has been insufficient recognition of the pluralistic heterogeneity of basic education resources itself, manifested in the lack of a classification of basic education resources and the insufficient consideration of preschool and high school education. Second, most studies have focused on the macro- and mesoscale. Research on microscale villages is lacking, especially on rural areas around large cities. Problems such as rural aging and labor loss have become acute. The rural exodus and the shrinkage of rural schools have negatively affected the normal educational resources accessed by some of the left-behind children. The overall unsatisfactory quality and accessibility of rural education is reflected in the spatial mismatch of educational resources. With the transformation and reconstruction of rural economies, basic public services have become a key to supporting residents' daily activities, quality of life and local economies, especially education, as an important way for members of society to accumulate cultural capital. Education constitutes one of the most important means to promote social mobility and reduce social differences. Therefore, it is essential to deepen the research on the diversified supply of and demand for rural basic education at the microscale.

Thus, we aim to reveal the spatial pattern of basic education resources and its influencing factors in rural areas on the periphery of large cities in China during the transition period, taking Wuhan as an example and basing the study on the village scale. It is hoped that this can provide a scientific basis for the study of the spatial allocation of basic rural education resources. This article is organized as follows (Figure 1). The Section 1 explains the significance and goal of studying rural basic educational resources. The Section 2 introduces the research area, data source and methodology. The Sections 3 and 4 analyzes the characteristics and influencing factors of the spatial distribution of rural basic educational resources in Wuhan's new urban districts. Based on the previous research, the Section 5 discusses the practical meaning of these findings and states the limitations of our research. The last Section 6 contains the conclusions.

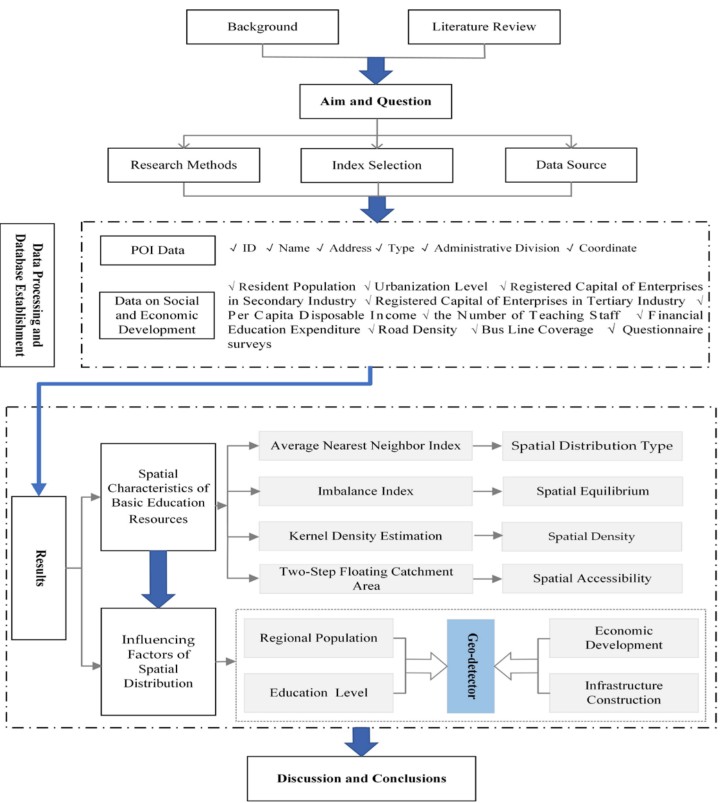

**Figure 1.** Research framework and steps.

## 2. Materials and Methods

### 2.1. Research Area

Wuhan is located in the center of the hinterland of China, specifically in the eastern Jianghan Plain and at the intersection of the Yangtze River and Hanjiang Rivers. It is also a central city in the central region and an important base for science and education in China. The research area is the rural area of six new urban districts under the jurisdiction of Wuhan City, namely, the Dongxihu District, Hannan District, Caidian District, Jiangxia District, Huangpi District and Xinzhou District, with a total area of 6636.94 km$^2$, a total of 81 streets (townships), and 1868 administrative villages (Figure 2). At the end of 2020, the permanent rural population in the new urban districts was 2.166 million, and the disposable income of permanent residents was 18,250 yuan. There are 602 basic education facilities in the new urban districts, including 277 kindergartens, 235 elementary schools and 90 secondary schools (including 7 high schools). The numbers of teaching staff are 4997, 5321 and 12,470, respectively, and the numbers of students enrolled are 39,819, 73,201 and 40,908, respectively. With regard to the scale of operation of rural basic education facilities in the new urban districts, there are significant differences in their existence and significant spatial differences in the educational services received by the school-age population. According to the proposal of the Wuhan municipal government, by 2025, the level of universal education in rural areas should be significantly improved, the education gap between urban and rural areas should be reduced, the foundation of high-quality development of rural education should be more solid, and the ability and level of education to serve rural revitalization should be improved.

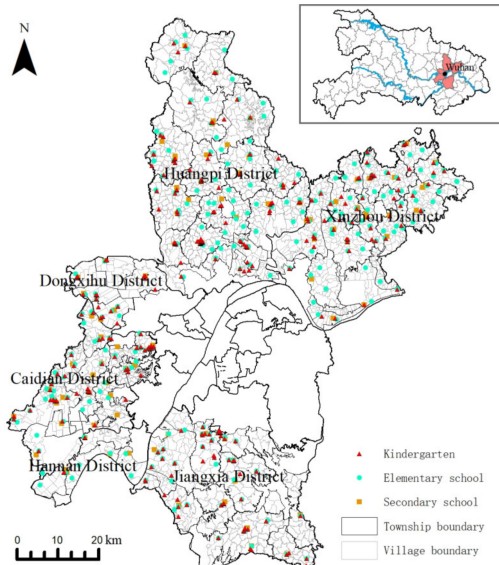

**Figure 2.** Location of the research area.

*2.2. Data Source*

The data used in this study mainly included POI data on basic education resources, data on basic geographic information and data on socioeconomic development and field surveys. The data details are as follows:

(1) POI data on basic education resources. This dataset was derived from the geographic service platform of Amap (https://lbs.amap.com/, accessed on 31 December 2020), including name, address, latitude, longitude, category, etc., and these were mainly obtained through web crawlers, screening and deduplication. As of 31 December 2020, we obtained POI data for a total of 602 basic education.

(2) Data on basic geographic conditions. The data on basic geographic conditions came from the Wuhan Geomatics Institute. There were a total number of 1868 administrative villages in Wuhan City's new urban districts in 2020. The data are stored in text and table formats and include information on location, land, population, etc. Data on the basic geographic information of Wuhan are kept in the format of ArcGIS vector data. The obtained data of the Wuhan map were geospatially matched with the data of rural basic education resources. In addition, road network data (highways, national highways, provincial highways, county highways, township highways and other roads) were downloaded from the Open Street Map website (http://www.openstreetmap.org, accessed on 31 December 2020), followed by topology checks to establish reasonable topological connectivity and construct road network datasets.

(3) Data on socioeconomic development and field surveys. Socioeconomic statistics were derived from the Wuhan Statistical Yearbook 2021, Wuhan Census of the Geographical Conditions Report 2021 and the Statistical Bulletin of National Economic and Social Development of Wuhan in 2021. In addition, the research team selected 48 administrative villages from September to November 2020, distributed 593 questionnaires and collected 576 valid questionnaires. Information related to the quality, service attitude, and economic distance of residents to educational public service facilities was obtained. In terms of distance (time), residents were asked if they expected their children to attend kindergartens, elementary schools and secondary schools within 10, 20 or 30 min of their residences.

*2.3. Research Methods*

According to Wuhan geographical conditions census data, this paper used ArcGIS to establish a geographic database of basic education; meanwhile, it used the nearest neighbor

index, imbalance index, kernel density analysis [51] and two-step floating catchment area to study the spatial distribution characteristics of rural basic education resources in Wuhan City's new urban districts and then used the geographic detector method [52,53] to analyze the influencing factors of its spatial distribution. The specific model and its geographical significance are shown in Table 1.

**Table 1.** Interpretation of basic geographic analysis model.

| Research Methods | Model Formula | Model Interpretation | Geographical Significance | Remarks |
|---|---|---|---|---|
| Nearest neighbor index | $R = \bar{r}_1 / \bar{r}_E$ $\bar{r}_E = \frac{1}{2\sqrt{n/A}}$ | $\bar{r}_1$ is the actual average nearest distance, $\bar{r}_E$ is the theoretical nearest distance, $n$ is the number of basic education facilities, and $A$ is the area of the new urban districts. | $R$ is the nearest neighbor index, if $R > 1$, $= 1$, $<1$, it means uniform, random, and clustered distribution, respectively. | Formula (1) |
| Imbalance index | $S = \frac{100 \sum_{i=1}^{n} Y_i - 50(n+1)}{100n - 50(n+1)}$ | $Y_i$ is the proportion of the number of various basic education facilities in the new urban districts to the total, according to the $i$-thcumulative percentage from largest to smallest, $n$ is the number of new urban districts. | $S$ is the imbalance index, if $S = 0$, it means uniform distribution, and if $S = 1$, it means the distribution is extremely uneven. | Formula (2) |
| Kernel density analysis | $f(x) = \frac{1}{nh} \sum_{i=1}^{n} K\left(\frac{x-x_i}{h}\right)$ | $K\left(\frac{x-x_i}{h}\right)$ is the kernel function, $x - x_i$ is the distance from the estimated point $x$ to event $x_i$, $h$ is the bandwidth, and $n$ is the number of basic education facilities within the threshold range. | $f(x)$ is the estimated density of basic education facilities at $x$, the larger the value, the denser the points, and the higher the probability of occurrence. | Formula (3) |
| Two-step floating catchment area method | $A_i^F = \sum_{j \in \{d_{ij} \leq d_0\}} \left( \frac{S_j}{\sum_{k \in \{d_{kj} \leq d_0\}} D_k} \right)$ | $d_{ij}$ is the distance between residential area $i$ and education point $j$, $D_k$ is the demand for the number of people in the search area (that is $d_{kj} \leq d_o$), $S_j$ is the total supply of point $j$, expressed by the number of teaching staff. | $A_i^F$ is accessibility, and the greater its value, the better accessibility. | Formula (4) |
| Geographic detector | $q = 1 - \frac{\sum_{h=1}^{L} N_h \sigma_h^2}{N\sigma^2}$ | $h = 1, 2, \ldots, L$ is the variable $Y$ or strata of factor $X$, $N_h$ and $N$ are the number of units in layer $h$ and the whole region respectively; $\sigma_h^2$ and $\sigma^2$ are are the variance of layer $h$ and $Y$ value of the whole region, respectively. | $q$ is the influence of each factor on the spatial distribution of basic education resources, and the value range is [0, 1]. The larger the value is, the greater the influence of the selected factor on the spatial distribution of basic education resources is; otherwise, the weaker it is. | Formula (5) |

## 3. Results

### 3.1. Spatial Distribution Types of Basic Education Resources

The results of the nearest neighbor index (Formula (1)) of rural basic education resources are shown in Table 2. The nearest neighbor indices of kindergartens and elementary schools were 0.54 and 0.82, respectively, both of which were less than 1 and passed the significance test, showing a clustering distribution. Therefore, the spatial correlation of these educational resources is high, which is conducive to the circulation and integration of educational resources. Among them, the nearest neighbor index of elementary schools was close to 1, which was a lightly clustered distribution; and the nearest neighbor point index of secondary schools was 1.07 > 1, which showed that secondary schools tend to be evenly distributed in spatial distribution. The main implication is that there are few secondary schools, they are mostly located in the township centers with a large service scope and the schools are far away from each other, making the average observed distance larger.

**Table 2.** The nearest neighbor index and its spatial distribution types of rural basic education resources in Wuhan City's new urban districts.

| Education Resources | Number | Theoretical Distance | Actual Distance | Nearest Neighbor Index | Type of Distribution | *p* Value |
|---|---|---|---|---|---|---|
| Kindergarten | 277 | 3147.078 | 1708.826 | 0.54 | clustered | 0.000 |
| Primary school | 235 | 3518.655 | 3235.339 | 0.82 | clustered | 0.010 |
| Secondary school | 90 | 6788.413 | 5318.159 | 1.07 | uniform | 0.000 |

### 3.2. Spatial Equilibrium Characteristics of Basic Education Resources

The imbalance index (Formula (2)) of rural basic educational resources was 0.36, indicating that basic education is unevenly distributed in the new urban districts. It is mainly concentrated in administrative villages in Huangpi District, Xinzhou District and Caidian District, and its percentage reached 75%. Specifically, the imbalance index of kindergartens was 0.32, mainly distributed in Huangpi District, Xinzhou District and Jiangxia District, with 86, 60 and 57, respectively, accounting for 73%. The imbalance index of elementary schools was 0.40, mainly distributed in Huangpi District, Xinzhou District and Caidian District, with the numbers 73, 70 and 41, respectively, accounting for 78%. The imbalance index of secondary schools was 0.30, mainly in Xinzhou District, Huangpi District and Caidian District, with 24, 22 and 22, respectively, accounting for 75% of the total (Table 3). According to the quantitative distribution characteristics of rural basic education resources in the new urban districts, the spatial distribution of rural basic educational resources is poorly balanced, tending to cluster in Huangpi District, Xinzhou District and Caidian District.

**Table 3.** The quantity distribution of rural basic educational resources in Wuhan City's new urban districts.

| District | Quantity and Percent of Basic Education Resources (Number, %) | | | | | | | | |
|---|---|---|---|---|---|---|---|---|---|
| | Kindergarten | Percent | Cumulative Percent | Primary School | Percent | Cumulative Percent | Secondary School | Percent | Cumulative Percent |
| Dongxihu | 21 | 8% | 8% | 9 | 4% | 4% | 6 | 7% | 7% |
| Hannan | 4 | 1% | 9% | 8 | 3% | 7% | 4 | 4% | 11% |
| Caidian | 49 | 18% | 27% | 41 | 17% | 25% | 22 | 24% | 36% |
| Jiangxia | 57 | 21% | 47% | 34 | 14% | 39% | 12 | 13% | 49% |
| Huangpi | 86 | 31% | 78% | 73 | 31% | 70% | 22 | 24% | 73% |
| Xinzhou | 60 | 22% | 100% | 70 | 30% | 100% | 24 | 27% | 100% |

### 3.3. Spatial Distribution Density of Basic Education Resources

By using the kernel density analysis tool in ArcGIS 10.2, a kernel density map of rural basic educational resources in the new urban districts was generated (Figure 3), and it was concluded that there were significant spatial differences in rural basic educational resources in the new urban districts. Among them, kindergartens formed three density core areas, which were mainly concentrated in Hengdian Street in the south of Huangpi District, Yongfeng Street in the east of Caidian District and Zhengdian Street in the north of Jiangxia District, with a distribution density between 2.69 and 4.47/10 km², while other areas were more scattered in their distributions. The four cluster density gathering areas formed by the elementary schools were mainly concentrated in Zhuru Street and Yongfeng Street in Caidian District, Zhengdian Street in the north of Jiangxia District, the junction of Hengdian Street and Fukou Street in the south of Huangpi District and Xinchong Town in the east of Xinzhou District, with a distribution density of 1.05–1.73/10 km². The secondary schools formed two density centers and five subdensity centers, which were mainly distributed at the junction of Zhuru Street in the west of Caidian District and Liji Street and Jiacheng Street in Xinzhou District, with a distribution density of 0.9–1.15/10 km². The distribution of secondary density centers was relatively scattered, and the density was between 0.47 and 0.92/10 km², of which a cluster agglomeration area was formed in the east of Caidian District. In general, the density of rural basic educational resources in the new urban districts presents a spatial distribution pattern of "block clustering and multicenter development".

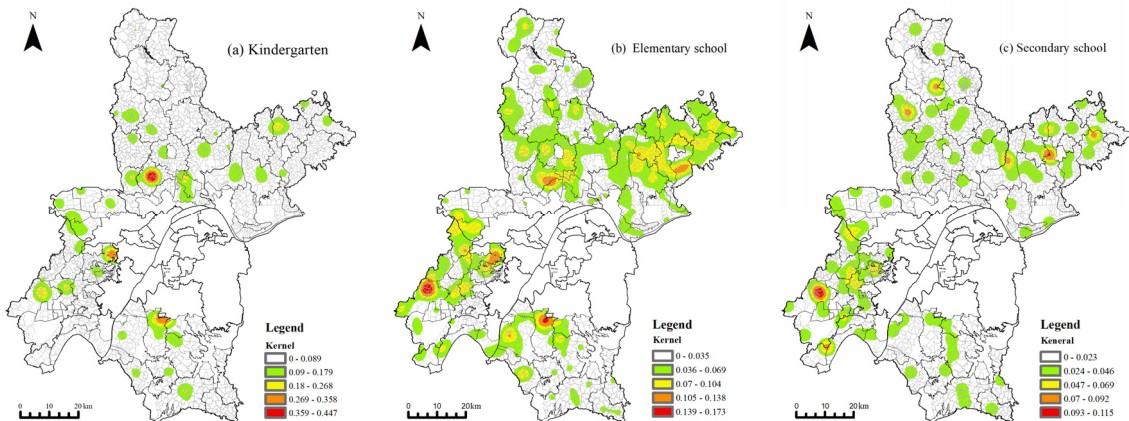

**Figure 3.** Kernel density analysis of rural basic education resources in Wuhan City's new urban districts.

### 3.4. Spatial Accessibility of Basic Education Resources

The spatial accessibility of rural basic education resources in Wuhan City's new urban districts was calculated and graded in Figure 4 and Table 4, indicating significant regional differences in the spatial accessibility of basic educational resources in the new urban districts. Specifically, the average value of kindergarten accessibility was 0.15, and the 1330 administrative villages below the average value (average and below) were widely distributed and accounted for 71.2%. Among them, 468 administrative villages had 0 accessibility, which is mainly attributed to these administrative villages being small in area, small in population, large in traditional farming area, backward in rural infrastructure, small in the distribution and scale of kindergartens, and affected by the distance attenuation effect and poor accessibility. The average value of elementary school accessibility was 0.08, and the 992 administrative villages below the average value (average and below) were mainly located in Huangpi District and Xinzhou District, which accounted for 53.1%. Among them, 30 administrative villages had an accessibility value of 0. Although these administrative villages are large in area and have a large number of elementary schools, the scale and service capacity of the elementary schools are limited. Moreover, county

and township roads are mainly distributed in these areas, with a single road and poor accessibility as a whole. The average value of the accessibility of secondary schools was 0.06, and the 857 administrative villages below the average value were mainly located on the periphery of the new urban districts, accounting for 45.9%. The accessibility of 22 administrative villages was 0, which is mainly attributed to the poor traffic conditions of these administrative villages. Furthermore, most of them do not have secondary school facilities, so residents need to spend more time reaching the nearest secondary school.

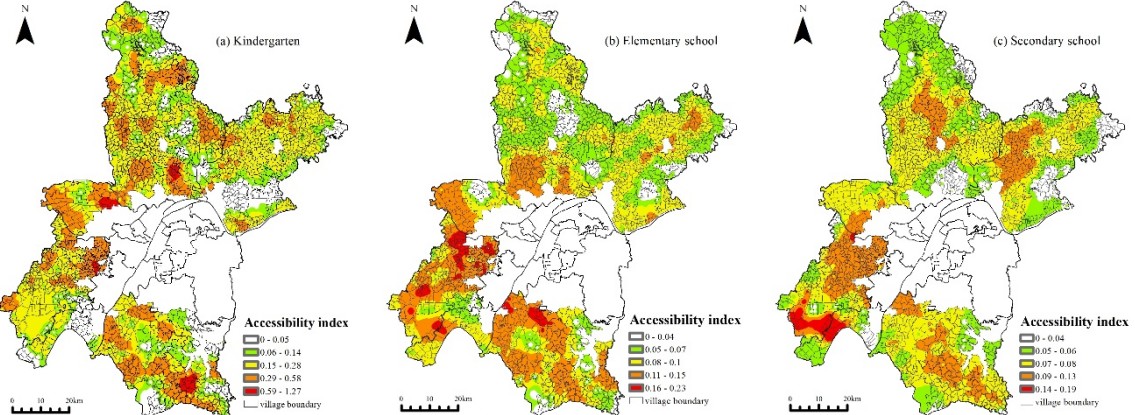

**Figure 4.** Spatial accessibility of rural basic education resources in Wuhan City's new urban districts.

**Table 4.** Index of accessibility of rural basic educational resources in Wuhan City's new urban districts.

| Level | Kindergarten | | | Primary School | | | Secondary School | | |
|---|---|---|---|---|---|---|---|---|---|
| | Accessibility Index | Villages (Number) | Percent (%) | Accessibility Index | Villages (Number) | Percent (%) | Accessibility Index | Villages (Number) | Percent (%) |
| Low | 0–0.05 | 665 | 35.60% | 0–0.04 | 263 | 14.08% | 0–0.04 | 394 | 21.09% |
| Relatively low | 0.06–0.14 | 625 | 33.46% | 0.05–0.07 | 534 | 28.59% | 0.05–0.06 | 463 | 24.79% |
| General | 0.15–0.28 | 447 | 23.93% | 0.08–0.1 | 511 | 27.36% | 0.07–0.08 | 521 | 27.89% |
| Relatively High | 0.29–0.58 | 119 | 6.37% | 0.11–0.15 | 456 | 24.41% | 0.09–0.13 | 470 | 25.16% |
| High | 0.59–1.27 | 12 | 0.64% | 0.16–0.23 | 104 | 5.57% | 0.14–0.19 | 20 | 1.07% |
| Average | 0.15 | | | 0.08 | | | 0.06 | | |
| Standard deviation | 0.12 | | | 0.04 | | | 0.03 | | |

The areas with high and higher accessibility of basic educational resources in the new urban districts were concentrated in Panlong and Wujiashan Street in Dongxihu District, Caidian Street in the middle east of Caidian District, Shamao Street in Hannan District and Zhifang Street in Jiangxia District. On the one hand, these areas are close to the central urban area and Chengguan town, with a dense population, fast urbanization process, relatively complete public service facilities and a high service level. On the other hand, these areas have the advantages of flat terrain, a high density of rural settlements, dense road networks and high road grades, thus reducing the time required for residents to reach basic educational facilities. In general, the accessibility of kindergartens showed a spatial pattern of "large dispersion and small clustering", with multiple high-value clustering areas, which are in close proximity to kindergartens. The accessibility of elementary and secondary schools showed a high spatial pattern in the south and a low spatial pattern in the north, mainly for two reasons. First, the supply and demand of the school-age population and basic education resources are relatively balanced in the south, and the road network is of a higher grade. Second, the preference of residents for schools with high-level service capacity has led some residents to "stay away from the near and seek the far" in the northern region, which has affected their spatial accessibility. For example, the administrative villages with the highest accessibility to elementary and secondary schools

are Taishan village in Caidian District and Laozihu village in Hannan District, respectively, and the basic educational service level in the two new urban districts is also high.

## 4. Influencing Factors of Spatial Characteristics

### 4.1. Selection of Influencing Factors

The spatial pattern of basic educational resources is the result of the combined effect of various socioeconomic factors. Combined with the relevant literature [54–56], nine indicators were selected for analysis from four aspects: regional population, regional economy, education development level and infrastructure construction. Among them, the regional population is characterized by the resident population ($X_1$) and urbanization level ($X_2$); the regional economy is characterized by the registered capital of enterprises in secondary industry ($X_3$), the registered capital of enterprises in tertiary industry ($X_4$) and per capita disposable income ($X_5$); the education development level is characterized by the number of teaching staff ($X_6$) and financial education expenditure ($X_7$); and the infrastructure construction is characterized by road density ($X_8$) and bus line coverage ($X_9$). It should be noted that since the ratio of construction land area partly reflects the urbanization level, the urbanization level is replaced by the ratio of the construction land area in each administrative village and the bus route coverage is replaced by the ratio of the bus stop buffer area (500 m) in each administrative village. Given that financial education expenditures are reflected in educational development as the campus land area, this indicator is used to reflect the financial education expenditure of each administrative village. Then, by discretizing each influencing factor according to the natural breakpoint method, the equal spacing method, the standard deviation method, and the K-means method, the above processing results were compared. The natural breakpoint method was chosen to classify each influencing factor into five categories, and each category was, in turn, enhanced. The contribution rate and interaction factors of each factor were quantitatively analyzed by using a geographical detector to analyze the influence mechanism of the spatial distribution of rural basic educational resources in Wuhan City's new urban districts.

### 4.2. Factor Detection

First, the linear relationships between the number of basic educational resources and the nine independent variables were analyzed using the Pearson correlation coefficient. Except for the registered capital of enterprises in the tertiary industry ($X_4$), there is a positive linear correlation between the other eight variables and the quantity of basic educational resources. The $q$ value of the factor detection result is the degree of explanation of the influence of the factor on the spatial distribution of basic educational resources. Eight variables passed the significance test, indicating that these eight variables were important drivers of the spatial distribution of rural basic educational resources in new urban districts (Table 5). Among them, the explanatory powers of the kindergarten influencing factor $q$ in descending order were as follows: number of residents (0.257) > urbanization level (0.181) > financial education expenditure (0.167) > road density (0.131) > per capita disposable income (0.106) > number of teaching staff (0.097) > secondary industry (0.083) > bus line coverage (0.005). The explanatory powers of the elementary school influencing factor $q$ in descending order were as follows: number of residents (0.246) > disposable income per capita (0.193) > financial expenditure on education (0.186) > number of teaching staff (0.137) > level of urbanization (0.130) > secondary industry (0.085) > road density (0.032) > bus route coverage (0.003). The ranking of the explanatory powers of $q$ of the secondary school influencing factors from largest to smallest were as follows: number of residents (0.219) > level of urbanization (0.196) > number of teaching staff (0.146) > disposable income per capita (0.133) > secondary industry (0.107) > financial education expenditure (0.085) > road density (0.025) > bus route coverage (0.006).

**Table 5.** Results of Pearson correlation coefficients and factor detector.

| Education Resources | Independent Variable | $X_1$ | $X_2$ | $X_3$ | $X_4$ | $X_5$ | $X_6$ | $X_7$ | $X_8$ | $X_9$ |
|---|---|---|---|---|---|---|---|---|---|---|
| Kindergarten | Pearson correlation coefficient | 0.720 ** | 0.637 ** | 0.534 ** | 0.321 | 0.619 * | 0.689 ** | 0.692 ** | 0.548 * | 0.375 * |
| | $q$ value of factor detection | 0.257 | 0.181 | 0.083 | 0.072 | 0.106 | 0.097 | 0.167 | 0.131 | 0.005 |
| | $p$ | 0.000 | 0.043 | 0.000 | 0.14 | 0.054 | 0.000 | 0.000 | 0.021 | 0.000 |
| Primary school | Pearson correlation coefficient | 0.732 ** | 0.623 ** | 0.588 ** | 0.244 | 0.627 * | 0.699 ** | 0.614 ** | 0.627 * | 0.339 * |
| | $q$ value of factor detection | 0.246 | 0.13 | 0.085 | 0.063 | 0.193 | 0.157 | 0.186 | 0.032 | 0.003 |
| | $p$ | 0.000 | 0.025 | 0.000 | 0.336 | 0.028 | 0.000 | 0.000 | 0.089 | 0.021 |
| Secondary school | Pearson correlation coefficient | 0.741 ** | 0.523 ** | 0.502 ** | 0.302 | 0.616 * | 0.809 ** | 0.671 ** | 0.630 * | 0.356 * |
| | $q$ value of factor detection | 0.219 | 0.196 | 0.107 | 0.118 | 0.133 | 0.146 | 0.085 | 0.025 | 0.006 |
| | $p$ | 0.000 | 0.064 | 0.000 | 0.365 | 0.085 | 0.000 | 0.000 | 0.039 | 0.000 |

* and ** are significant at 10% and 5% levels, respectively.

According to the above rankings of $q$ values, the explanatory power of the resident population ($X_1$) on the spatial distribution of each type of basic educational resource was the largest (0.257, 0.246 and 0.219, respectively), indicating that it is the most important factor influencing the spatial distribution of basic educational resources in the rural areas of the new urban districts. The essence of the spatial distribution of basic educational resources is the dynamic match between the supply of educational resources and the demand of the population in a mutually organic combination, with the fundamental purpose of education being to promote the all-around development of society. The larger the population in a region, the greater the demand for education, and therefore the easier it is to attract the clustering of talent and resources, which is of great significance for the agglomeration development of educational resources.

The explanatory power of the level of urbanization ($X_2$) was secondary school> kindergarten > elementary school, second only to the resident population. For the allocation of rural basic education resources, the level of urbanization development has a positive driving effect. Secondary schools are generally located in or around township centers with a high level of urbanization due to their high scale and large service scope. Kindergarten is the initial stage of basic education, and as a quasi-public good, it has strong positive externalities. On the one hand, it requires the financial investment of the government, and on the other hand, the opening of the national education market attracts all kinds of private capital. Kindergartens, as providers of preschool education, are greatly regulated by the market, so educational resources tend to flow to those areas with a larger population and better urban development conditions. Kindergarten density also showed that its core areas were mainly distributed at the junction of the new urban districts and the central urban area and the streets with good development conditions in each district, indicating that the resources of the central urban area radiate out toward these areas.

The explanatory power of the secondary industry enterprise registered capital ($X_3$) was secondary school > elementary school > kindergarten. Given that industry is the core foundation that supports the economic development of villages and is also important for the development and growth of the economy, it is possible to consider improving local education, health care and public services. Secondary schools, the most important level of compulsory education, have a large population to serve. Therefore, they are generally located in areas with good economic conditions in the countryside, and their development is more dependent on the overall economic level and support of society than kindergartens.

The explanatory power of disposable income per capita ($X_5$) was elementary school > secondary school> kindergarten, with disposable income per capita representing the average economic freedom and standard of living of residents. When the disposable income of the entire population is higher, the likelihood of having elementary school resources in their area is higher. With the implementation of the policy of equalizing basic public services, parents want their children to receive higher-quality educational resources; thus, they go to areas with concentrated educational resources and good accessibility, that is, the areas with high accessibility and good socioeconomic development conditions, as shown

in the above analysis (the eastern part of Dongxihu District, the central-eastern part of Caidian District and the northern part of Jiangxia District). Moreover, as villagers go out to work, they can pay higher education and living costs.

The explanatory power of the number of teaching staff ($X_6$) was secondary school > elementary school > kindergarten. Secondary school, as the final stage of compulsory education, is also the most important stage for parents. The quality of its educational resources is directly related to the future of students, and the teaching staff is the key element of secondary education in this regard. Good teaching staff not only expands the influence of the school but also helps the radiation and spread of secondary education resources (such as building new secondary schools) and influences the spatial distribution of secondary education resources, reflecting the importance of the "people" factor in shaping the distribution of secondary education resources.

The explanatory power of fiscal education expenditure ($X_7$) was elementary school > kindergarten > secondary school. Since public welfare is the fundamental attribute of basic education, the local government is the main source of educational resources. Therefore, the financial expenditure capacity of local governments has an important influence on the allocation level of rural educational resources. Given that there are significant differences in the scale and intensity of local governments' investment in promoting rural basic educational construction, it affects the spatial distribution of elementary schools and kindergartens.

The influence of road density ($X_8$) and bus route coverage ($X_9$) on the spatial distribution of basic educational resources was generally weak. Road density had some influence on the distribution of kindergartens. This is mainly because the age of kindergarten children is generally younger, so families are more sensitive to the "home-to-school" commuting distance, and the density of road networks in urban areas is relatively high. The correlation between urbanization and the spatial distribution of kindergartens is also confirmed in this study. In addition, the insignificant effect of the registered capital of tertiary sector enterprises ($X_4$) on the spatial distribution of various types of basic education resources indicates that the mutual gain effect of the tertiary sector and education has not yet been identified.

*4.3. Interaction Detection*

Interaction detection is used to detect whether the influence of two factors acting together on the spatial distribution of basic education resources increases or decreases. The results showed that the interactions of any two of the eight influencing factors are of two types, nonlinear enhanced and double-factor enhanced, of which the nonlinear enhanced is predominant. This suggests that the interaction of any two influencing factors increased the explanatory power of the spatial differentiation of basic educational resources (Table 6). Therefore, the spatial distribution of basic educational resources in Wuhan City's new urban districts is actually the result of the combined effect of multiple factors. In kindergartens, the interaction between the resident population and urbanization was the strongest (0.474), followed by the registered capital of secondary industry enterprises (0.356) and financial education expenditure (0.354). In elementary schools, the interaction between the number of residents and urbanization was the strongest (0.362), followed by disposable income per capita (0.352) and financial education expenditure (0.309). In secondary schools, the interaction between the resident population and urbanization was the strongest (0.414), followed by the number of teaching staff (0.365) and the registered capital of secondary industry enterprises (0.354). Regarding the interactions between the other factors, since they were all weaker than the interactions between the size of the resident population and the other seven factors, the size of the resident population combined with the other factors was the most influential factor in the spatial distribution of rural basic educational resources in Wuhan City's new urban districts. This further indicated that the spatial distribution of basic education in the new urban districts essentially reflected the difference in the number of permanent residents in each district and depended on the level of socioeconomic development. Districts with large populations and good socioeconomic conditions will

continue to be agglomerations of educational resources in the future, while those with small populations and poor socioeconomic conditions may face greater educational disparities.

**Table 6.** The results of interaction detection.

| Basic Education | Interaction | $X_1$ | $X_2$ | $X_3$ | $X_4$ | $X_5$ | $X_6$ | $X_7$ | $X_8$ | $X_9$ |
|---|---|---|---|---|---|---|---|---|---|---|
| Kindergarten | $X_1$ | 0.257 | | | | | | | | |
| | $X_2$ | 0.474 | 0.181 | | | | | | | |
| | $X_3$ | 0.356 | 0.414 | 0.083 | | | | | | |
| | $X_4$ | 0.272 | 0.106 | 0.190 | 0.072 | | | | | |
| | $X_5$ | 0.261 | 0.204 | 0.187 | 0.206 | 0.106 | | | | |
| | $X_6$ | 0.260 | 0.390 | 0.189 | 0.185 | 0.182 | 0.097 | | | |
| | $X_7$ | 0.354 | 0.298 | 0.237 | 0.186 | 0.183 | 0.173 | 0.167 | | |
| | $X_8$ | 0.270 | 0.206 | 0.197 | 0.137 | 0.216 | 0.191 | 0.195 | 0.131 | |
| | $X_9$ | 0.282 | 0.116 | 0.097 | 0.108 | 0.108 | 0.104 | 0.185 | 0.192 | 0.005 |
| Elementary school | $X_1$ | 0.246 | | | | | | | | |
| | $X_2$ | 0.362 | 0.130 | | | | | | | |
| | $X_3$ | 0.267 | 0.369 | 0.085 | | | | | | |
| | $X_4$ | 0.157 | 0.210 | 0.149 | 0.063 | | | | | |
| | $X_5$ | 0.352 | 0.311 | 0. 195 | 0.195 | 0.193 | | | | |
| | $X_6$ | 0.264 | 0.260 | 0.158 | 0.158 | 0.258 | 0.157 | | | |
| | $X_7$ | 0.309 | 0.340 | 0.256 | 0.226 | 0.272 | 0.266 | 0.186 | | |
| | $X_8$ | 0.278 | 0.254 | 0.166 | 0.109 | 0.196 | 0.198 | 0.191 | 0.032 | |
| | $X_9$ | 0.269 | 0.250 | 0.090 | 0.085 | 0.195 | 0.163 | 0.190 | 0.042 | 0.003 |
| Secondary school | $X_1$ | 0.219 | | | | | | | | |
| | $X_2$ | 0.415 | 0.196 | | | | | | | |
| | $X_3$ | 0.326 | 0.303 | 0.107 | | | | | | |
| | $X_4$ | 0.231 | 0.204 | 0.183 | 0.118 | | | | | |
| | $X_5$ | 0.224 | 0.206 | 0.143 | 0.136 | 0.133 | | | | |
| | $X_6$ | 0.365 | 0.342 | 0.234 | 0.165 | 0.262 | 0.146 | | | |
| | $X_7$ | 0.288 | 0.214 | 0.182 | 0.126 | 0.190 | 0.231 | 0.085 | | |
| | $X_8$ | 0.238 | 0.221 | 0.139 | 0.136 | 0.165 | 0.175 | 0.117 | 0.025 | |
| | $X_9$ | 0.225 | 0.210 | 0.116 | 0.120 | 0.139 | 0.170 | 0.112 | 0.038 | 0.006 |

## 5. Discussion

*5.1. Influencing Factors of Existing Differences in Different Types of Basic Education Resources*

Education equity is a goal of modern society. The quantity, quality and richness of the allocation of rural basic education resources directly affect the quality of life of rural residents and whether rural society can achieve quality-oriented management and people-oriented modern social governance. The allocation and spatial differentiation of educational resources is an important component of geographic research. One of the important issues in the development of China's new urbanization is to promote the equalization of public services in basic education, which is shaped by factors related to population, economic development and residents' income, as well as historical dimensions. Inequality in educational facilities will continue to exist in the short term in China. Therefore, the quantitative evaluation of educational facilities has great theoretical and practical significance and can provide a reference for the development of education equity policy. By exploring the spatial pattern of rural basic educational resources and their influencing factors in the periphery of large cities at the village scale, this paper concludes that the spatial distribution pattern of rural basic educational resources in Wuhan City's new urban districts can be better explained, which has important practical value for optimizing the spatial distribution of basic educational resources and rural revitalization. Furthermore, to scientifically determine the spatial pattern of basic education resources, it is necessary to pay more attention to the main influencing factors derived from the adjustment analysis. Given the differences in the influencing factors for the formation of the spatial patterns of basic educational resources in different regions and types, it is indispensable to understand that various factors work

together to balance the spatial pattern of basic educational resources and gradually achieve equalization by adapting to local conditions.

### 5.2. Measures to Optimize the Configuration of Rural Basic Education Resources

With the widespread popularization of compulsory education, rural compulsory education plays an increasingly important role in promoting the all-around development of rural agriculture and increasing farmers' income. China has long been influenced by emphasizing the city over rural areas, and the study of rural public service facilities (including basic education) has always lagged behind that of urban public service facilities. It is difficult for the government-centered rural basic education supply model [57] to cope with the funding demand for basic education resources in the rural areas of Wuhan. With the improvement of social and economic levels, rural residents' quality of life is constantly improving, and they have greater demands for a better quality of life. Additionally, the rapid development of the internet has impacted the traditional way of life of rural residents and increased public awareness and demand for human services. In other words, in the study of rural basic education resources, demand is much stronger than supply. The main body responsible for the configuration of rural basic education facilities should coordinate relevant standard requirements for the participation of various departments in their construction, provide guidance in an orderly manner, implement strict inspections and do a good job at guaranteeing the configuration and construction of rural basic education from government departments. First, in the process of the development of rural industries and the gradual increase in economic agents, market forces should be actively introduced or public–private partnerships should be used to increase the supply of rural basic education facilities and relieve the pressure of insufficiently allocated funds. Second, the upgrading of industrial structures and the urbanization level in the new urban districts should be steadily promoted, farmers' income should be increased through multiple channels and an upward cycle of "income growth-improvement of education level" should be formed. Finally, an orderly and open teacher training organization system should be established to improve the hierarchical relationship in the training system by formulating systems and policies. At the same time, commercial organizations, nongovernmental nonprofit organizations and trusted international organizations should actively contribute to teacher training in the rural compulsory education stage.

### 5.3. Strengths, Limitations and Prospects

The configuration of rural basic education resources is subject to the interaction of internal foundations and the external environment. In this study, we focused on the quantity and type of rural basic education resources supplied to residents. First, we analyzed the spatial distribution type, equilibrium and density of educational resources, and then we calculated the distance between rural settlements and educational facilities based on real rural road networks to objectively reflect the actual range rural residents must travel. Second, the differences in influencing factors of different types of basic education facilities were considered; these had different effects on facility usability, and more attention was given to the differentiated factors to compensate for the shortcomings of previous studies on the same topic. Third, we explored the distribution and influencing factors of rural basic education resources from a micro perspective, which helps guide the allocation and optimization of public education facilities in rural areas in the context of rapid industrialization and integrated urban–rural development and provides a theoretical basis for China's rural revitalization strategy.

The allocation and spatial differentiation of educational resources is an important research direction of educational geography, and the results provide some guidance and suggestions for rural basic education. Regarding future studies, this paper still needs further exploration and improvement in the following aspects. First, since the study used only annual data to analyze the spatial pattern of basic education resources and the influencing factors in Wuhan City's new urban districts, it did not reflect the patterns and trends of its

evolution over time. Thus, it is suggested that future studies combine data from multiple years, add the time dimension and explore the spatial and temporal distribution of basic education from the perspective of temporal evolution. Second, for the public service of basic education, it is necessary to consider both the quantity and the quality of education resources, because the unequal quality of education triggers residential migration and thus intensifies urban–rural sociospatial reconfiguration, reshaping the urban–rural map. The spatial distribution of high-quality educational resources should be focused on to reflect the current relationship between educational development and urban–rural social problems. Finally, MAUP (modifiable areal unit problem) will affect the results of spatial statistical analysis due to the differences between research units and divided regions. In future research, comprehensive analysis can be carried out under a variety of scales and zoning schemes, so that the research conclusion is closer to reality.

## 6. Conclusions

The main conclusions of the above analysis of the spatial distribution and influencing factors of rural basic education resources in Wuhan City's new urban districts based on GIS and geographic detectors are as follows.

(1) With regard to the type of spatial distribution, rural kindergartens and elementary schools showed a clustered distribution pattern, while secondary schools showed a uniform distribution pattern. The spatial distribution of rural basic educational resources was poorly balanced, tending to cluster in Huangpi District, Xinzhou District and Caidian District. With regard to the density of spatial distribution, the spatial distribution of rural basic educational resources varied significantly, showing the overall spatial distribution characteristics of "block clustering and multicenter development".

(2) The spatial accessibility of rural basic educational resources in the new urban districts is generally poor, and most of the villages had lower accessibility values than the average values for kindergartens, elementary schools and secondary schools, with significant regional differences. The spatial accessibility of kindergartens showed a spatial pattern of "large dispersion and small clustering", with multiple high-value clusters, which is related to nearby kindergarten enrollment. The accessibility of elementary schools and secondary schools showed a spatial pattern of high in the south and low in the north, which is mainly attributed to the supply of basic education resources and the behavioral preferences of residents.

(3) The main factors influencing the distribution of rural basic educational resources in the new urban districts include population, economy and education development levels. The influence of infrastructure construction is weak, and the core factors of the spatial distribution of each type of basic educational resource exhibit both consistency and some differences. The core influencing factors of kindergartens are, in order, the resident population, urbanization, secondary industry and financial education expenditure; the core influencing factors of elementary schools are, in order, of resident population, per capita disposable income, financial education expenditure and secondary industry; and the core influencing factors of secondary schools are, in order, the resident population, urbanization, number of teaching staff and secondary industry. The interactions of all independent variable indicators are nonlinear-enhanced or double-factor-enhanced, indicating that the joint effects among factors have a greater influence on the spatial distribution of basic education resources than the effect of any single factor.

Based on the above conclusions, the following suggestions are put forward: ① Establishing a flexible financial management system for compulsory education to fully release the dividends of supply-side reform. The study found that the basic education resources in the north are higher than those in the south and the west is higher than those in the east, and the distribution of all types of basic education facilities is not balanced. This is mainly due to the differences in social economy and population size. With the continuous progress of new urbanization in Wuhan, the trend of population flow to urban areas is further

strengthened, and the mechanical migration of population will have a more significant impact on the regional population size and structure. In this case, the original "two main" compulsory education management system exposed the lack of flexibility, and it is difficult to properly solve the problem of migrant children enrollment limitations. Therefore, government should adhere to the people-oriented new urbanization and establish a flexible financial management system for compulsory education. ② Supporting educational facilities flexibly according to the population structure and demand difference of different regions. On the one hand, when selecting and constructing basic educational facilities, attention should be paid to reducing the overlap of service radius, improving the efficiency of resource utilization and promoting the accessibility of educational resources. At the same time, the scale of educational facilities should gradually transition from the "urban–rural" two levels to the more reasonable "urban–town–village" three levels, promoting the coordinated development of educational resources. ③ The government should increase the proportion of high-quality teachers continuously in rural areas and incline policies such as talent introduction and capital investment to rural areas, and focus on teacher training. Then, we should improve school hardware facilities to promote the balanced development of rural schools, build an efficient communication platform between urban schools and rural schools, promote the interconnection of high-quality education resources and ensure the all-round quality balance of compulsory education opportunities, processes, quality and results.

**Author Contributions:** This paper was written with the contribution of all authors as follows: Conceptualization, Liang Jiang; Data curation, Ye Tian and Liang Jiang; Funding acquisition, Jing Luo and Ye Tian; Investigation, Liang Jiang, Ye Tian and Jie Chen ; Methodology, Jie Chen and Liang Jiang; Project administration, Jing Luo; Supervision, Jing Luo; Visualization, Liang Jiang; Writing—original draft, Liang Jiang and Jie Chen ; Writing—review & editing, Jie Chen and Jing Luo. All authors have read and agreed to the published version of the manuscript.

**Funding:** This research was supported by the National Natural Science Foundation of China (No. 41871176, No. 42001185, No. 42271228), the Humanities and Social Science Foundation of Ministry of Education of China (No. 20YJCZH147).

**Institutional Review Board Statement:** Not applicable.

**Informed Consent Statement:** Not applicable.

**Data Availability Statement:** Publicly available dataset was analyzed in this study. The POI data of education resources was available online from https://lbs.amap.com/ (accessed on 31 December 2020).

**Acknowledgments:** Thanks to the Wuhan Geomatics Institute (WGI) for contributions during data collection. Meanwhile, we are grateful for the comments and contributions of the anonymous reviewers and the members of the editorial team.

**Conflicts of Interest:** The authors declare no conflict of interests.

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
