# Peer review of "Spatial Pattern and Influencing Factors of Basic Education Resources in Rural Areas around Metropolises—A Case Study of Wuhan City’s New Urban Districts"

_ijgi, doi:10.3390/ijgi11110576_

Round 1
Reviewer 1 Report
This is an excellent work that deals with a very important issue such as the provision of educational infrastructure in rural areas.
A possible problem stems from the apparent failure to consider that schools may have different capacities. An area with three schools in close proximity may appear to be better endowed than another with only one school but three times as large.
I do not know if in China, the schools are very similar in terms of size, which would eliminate this problem.
The technical aspects are treated with care, and the presentation of the results is clear.
I believe it can be published with minor modifications. Some reference should be made to the issue of possible size differences in schools and perhaps requires some revision of the English.
Author Response
Dear Editor and Reviewers,
Thank you for your letter and for the reviewers’ comments concerning our manuscript. Those comments are all valuable and helpful for revising and improving our paper.
We are uploading (a) our point-by-point response to the comments (below) (response to reviewers), (b) an updated manuscript with highlighting indicating changes.
Best regards,
Liang Jiang, Jie Chen, Ye Tian, Jing Luo
Point 1: This is an excellent work that deals with a very important issue such as the provision of educational infrastructure in rural areas.
Response 1: Thanks for your comment.
As an important carrier of popularizing basic education, the spatial distribution and driving factors of basic education have become important topic in the field of public service facilities. Rural education is the endogenous power of rural revitalization. Under the dual background of unbalanced urban and rural development and rural revitalization in China, on the one hand, the development of rural basic education still faces many difficulties. On the other hand, the rural revitalization strategy not only indicate the direction of rural economic revitalization in China, but also guide the rural local government to issue relevant policies to revitalize basic education and promote the development of rural basic education. So, this paper studies the spatial pattern and influencing factors of rural basic education resources in Wuhan.
Point 2: A possible problem stems from the apparent failure to consider that schools may have different capacities. An area with three schools in close proximity may appear to be better endowed than another with only one school but three times as large. I do not know if in China, the schools are very similar in terms of size, which would eliminate this problem.
Response 2: Thanks for your comment.
School variables, such as school size and school location, matter. China has implemented a long-term dual economic system with a large number of high-quality educational resources concentrated in cities, while the resources of educational facilities and teachers in rural areas are obviously lower than those in cities. The educational resources in cities are superior to those in rural areas, and there is a big difference in the size of schools. In urban areas, the spatial distribution of basic education resources is different in quantity and quality. Under the “urban priority and urban-oriented” urban-rural relationship, Chinese urban education and rural education have embarked on an “urban priority and urban-oriented” path that has resulted in increasing disparities between the rural and urban education systems. With the continuous advancement of China's new urbanization process, the trend of population migration to urban areas has been further strengthened. The imbalance of educational resources led to residential migration, which has further intensified the reconstruction of urban and rural social space and reshaped the urban and rural landscape. For rural areas, although compulsory education has basically been fully covered, there is a large space in rural areas; In terms of population structure, the rural population accounts for a large proportion, and the school-age population requiring basic education is large; the level of economic development is relatively lagging behind, with low per capita income, limited fiscal revenue and insufficient investment in education; In terms of setting up primary and secondary schools, there are many teaching points, but the educational facilities are backward, the overall level of teachers is low, and the school size is not large but similar. In addition, the number of migrant children receiving education in cities is gradually increasing, resulting in a general shortage of students in rural primary and secondary schools, and many rural basic education is shrinking. Therefore, the size of schools in rural areas is relatively similar.
We agree that an area with three schools in close proximity may appear to be better endowed than another with only one school but three times as large. However, there is no big difference in the quality of education received for school-age population in rural areas, and more is the improvement of accessibility.
Point 3: The technical aspects are treated with care, and the presentation of the results is clear.
I believe it can be published with minor modifications. Some reference should be made to the issue of possible size differences in schools and perhaps requires some revision of the English.
Response 3: Thanks for your comment. We have added the size differences in schools in the revision as follows:
Lines 53-61:Although Chinese government started the Rural School Mapping Adjustment (RSMA) by withdrawing and merging most village-level schools that were considered to be inefficient and costly, promoting boarding schools for rural basic education, shifting primary schools from the village to the township and shifting secondary schools from the town-ship to the county[9], this actually makes the size of county and town schools show a trend of expansion, and the basic education in rural areas shrinks, exacerbating the equality and efficiency issues caused by the expansion of school size in the context of rural school layout adjustment[10,11]. It should be emphasized that educational equality can help provide educational resources to every student [12].
According to the reviewer’s comments, we have reorganized the structure of the article carefully and tried to avoid grammar or syntax error in the revision. In addition, our manuscript has been revised by American Journal Experts (AJE).

Reviewer 2 Report
The manuscript presents an application to evaluate the spatial pattern and influencing factors concerning education resources in the area around Wuhan city.
The application of indices is clearly presented and the extensive description of data and methods, such as dispersion/cluestering evaluation, etc., and the discussion of the results enable the reader to completely follow up the work that was done.
I feel, however, that the application of these indices is not new for the geo-information community, and the manuscript does not provide more than a straightforward application/calculation (and discussion of the obtained values). For instance, all distances are Euclidean and not real travel distances that could be calculated using a GIS. The effects of the MAUP were not considered or at least discussed, as there are calculations using administrative spatial units.
I recommend the authors to at least discuss the adherence to reality of their results using more sophisticated spatial modelling, namely the consideration of more plausible distances for the accessibility indicator.
To sum up, the article is very written and clear, but the novelty and interest to the geo-information community is no more than average, due to the absence of spatial reasoning beyond the analysis of point distributions with well-known indices in a 2D space.
Author Response
Dear Editor and Reviewers,
Thank you for your letter and for the reviewers’ comments concerning our manuscript. Those comments are all valuable and helpful for revising and improving our paper.
We are uploading (a) our point-by-point response to the comments (below) (response to reviewers), (b) an updated manuscript with highlighting indicating changes.
Best regards,
Liang Jiang, Jie Chen, Ye Tian, Jing Luo
Point 1: The manuscript presents an application to evaluate the spatial pattern and influencing factors concerning education resources in the area around Wuhan city. The application of indices is clearly presented and the extensive description of data and methods, such as dispersion/cluestering evaluation, etc., and the discussion of the results enable the reader to completely follow up the work that was done.
Response 1: Thanks for your comment.
As an important carrier of popularizing basic education, the spatial distribution and driving factors of basic education have become important topic in the field of public service facilities. Rural education is the endogenous power of rural revitalization. Scientific and rational allocation of educational resources is the focus of building a basic public education service system with balanced distribution, and is also a necessary measure to coordinate the development of educational facilities with the natural space of the city and social economy. This paper provides a methodological framework to measure and analyze educational resource allocation within and across systems and ascertain potential equity implications. Moreover, this work can enrich the research on the spatial distribution of rural compulsory education resources, and provide reference for the optimization and balanced layout of educational resources in Wuhan.
Point 2: I feel, however, that the application of these indices is not new for the geo-information community, and the manuscript does not provide more than a straightforward application/calculation (and discussion of the obtained values). For instance, all distances are Euclidean and not real travel distances that could be calculated using a GIS. The effects of the MAUP were not considered or at least discussed, as there are calculations using administrative spatial units.
Response 2: Thanks for your comment.
With the change of urban spatial structure and population size, the construction of rural compulsory education facilities lags behind, it is valuable to focus on the spatial pattern and equity of educational resources in rural areas. Scholars have explored the spatial pattern and equity of educational resources from different perspectives, but these studies have limitations.
The current studies on educational issues have mainly been statistical analyses and economic measurements. Research on the analysis of education resources using a geographic information system (GIS) spatial analysis and spatial econometric models is scant. In terms of data, research on education has mainly used statistical data from a district or several schools in the city as samples. However, these methods and data only analyse the fairness of educational resources from the macro scale of supply and demand, which is difficult to effectively reflect the real situation. Geographical information system and spatial analysis are effective ways to reveal the spatial pattern of educational resources and promote equity in education. As such, the convergence of multi-source data such as school POI, internal data, and socioeconomic data can provide a more comprehensive analysis of the specifics of spatial distribution of educational resources. This has important implications for exploring differences in the equity of educational resources between regions (i.e., urban areas, urban fringe areas, and rural areas).
All distances are Network Distance and real travel distances that could be calculated using a GIS. Road network data (highways, national highways, provincial highways, county highways, township highways and other roads) were downloaded from the Open Street Map website (http://www.openstreetmap.org), followed by topology checks to establish reasonable topological connectivity and construct road network datasets. In addition, the data of Wuhan traffic roads from Wuhan Geomatics Institute were obtained for supplement, and the distance from residential point to school was calculated using O-D cost matrix under network analysis in ArcGIS.
China’s educational resources have a strong administrative attribute, that is, the educational resources in each area are affected by the local government. On the one hand, the government uses the household registration system to decide the configuration of educational resources, on the other hand, the central government has shifted the administrative power of rural basic education. The spatial imbalance of basic education resources is deeply affected by China's educational territorial management system and administrative resources. So this paper takes the administrative village as the space unit. Due to different political and economic environments in different regions, the standards for the optimal allocation of public education resources are different. Our study only covered the area of Wuhan due to the limitation of data. Given that the education equality in rural areas around metropolises of China must be addressed, it is necessary to conduct future studies on the distribution of educational resources in other rural areas around metropolises in China. MAUP affect the results of spatial statistical analysis due to the differences between research units and divided regions. In this study, the administrative village is taken as the spatial unit, and the criteria proposed to evaluate the spatial mismatch of Wuhan's education resources are applicable to a certain stage of development of a specific research unit. In future research, comprehensive analysis can be carried out under a variety of scales and zoning schemes, so that the research conclusion is closer to reality.
Point 3: I recommend the authors to at least discuss the adherence to reality of their results using more sophisticated spatial modelling, namely the consideration of more plausible distances for the accessibility indicator.
To sum up, the article is very written and clear, but the novelty and interest to the geo-information community is no more than average, due to the absence of spatial reasoning beyond the analysis of point distributions with well-known indices in a 2D space.
Response 3: Thanks for your comment.
In the context of China’s new urbanization, the coordinated promotion of rural revitalization is essential to achieve urban and rural coprosperity and to promote modernization. The equitable development of education provides the foundation for the new urbanization. At present, research on basic education is concentrated in cities. Along with the continuous migration of young and middle-aged people from rural areas to cities and towns, a decrease in the population of school-aged children has led to the consolidation of schools for teaching, and schools have shown spatial scarcity in rural areas. This paper selects education POI data, which have the advantage of larger sample sizes and more meticulous information coverage. In addition to the social and economic data of administrative villages and questionnaire survey data, combined with GIS spatial analysis model, the accuracy and real-time performance of data spatial research and analysis can be effectively improved. Taking Wuhan as an example, this paper studies the spatial distribution and influencing factors of rural basic education resources, these findings will help governments in China and other countries further optimize the allocation of educational facilities in similar geographical areas.
The innovation of this study lies in the richness of data types and the number of spatial measurement samples. It combines spatial visualization and spatial measurement to explore the spatial distribution and influencing factors of different types of basic education resources and provide a new perspective for the research on education equality in other rural areas.
The distance threshold in this paper is based on the questionnaire survey of rural residents, which can basically reflect their distance demand for educational resources, and the results are relatively reliable. In the future analysis, on the one hand, more sophisticated spatial modelling should be selected to reveal the mechanism effects of different types of educational resources distribution. On the other hand, the combination of a questionnaire survey and spatial analysis, sociological and spatial geography, and spatial changes in students’ school choice behavior, along with county urbanization, should be explored.

Reviewer 3 Report
The paper analyzed the spatial distribution characteristics of rural basic education resources based on the methods of the average nearest neighbor index, imbalance index, kernel density analysis and two-step floating catchment area, and then used geographic detector analysis to detect its influencing factors in Wuhan City. The paper is well written and well structured. Methods and obtained results are well described. However, the authors may discuss in more detail the implications of the obtained results of the spatial distribution characteristics and possible measures to be taken by the authorities to improve education accessibility in rural areas if needed.
The references in the introduction should be in brackets.
The quality of Figure 4 should be improved and possibly put it in color like Figure 3.
Author Response
Dear Editor and Reviewers,
Thank you for your letter and for the reviewers’ comments concerning our manuscript. Those comments are all valuable and helpful for revising and improving our paper.
We are uploading (a) our point-by-point response to the comments (below) (response to reviewers), (b) an updated manuscript with highlighting indicating changes.
Best regards,
Liang Jiang, Jie Chen, Ye Tian, Jing Luo
Point 1: The paper analyzed the spatial distribution characteristics of rural basic education resources based on the methods of the average nearest neighbor index, imbalance index, kernel density analysis and two-step floating catchment area, and then used geographic detector analysis to detect its influencing factors in Wuhan City. The paper is well written and well structured. Methods and obtained results are well described.
Response 1: Thanks for your comment.
As an important carrier of popularizing basic education, the spatial distribution and driving factors of basic education have become important topic in the field of public service facilities. Rural education is the endogenous power of rural revitalization. Under the dual background of unbalanced urban and rural development and rural revitalization in China, on the one hand, the development of rural basic education still faces many difficulties. On the other hand, the rural revitalization strategy can not only indicate the direction of rural economic revitalization in China, but also guide the rural local government to issue relevant policies to revitalize basic education and promote the development of rural basic education. So, this paper studies the spatial pattern and influencing factors of rural basic education resources in Wuhan.
Point 2: However, the authors may discuss in more detail the implications of the obtained results of the spatial distribution characteristics and possible measures to be taken by the authorities to improve education accessibility in rural areas if needed.
Response 2: Thanks for your comment.
The implications of the obtained results of the spatial distribution characteristics are as follows. From the perspective of urban economics, the spatial mismatch of educational resources that deviates from the optimal allocation state in urban and rural development has resulted in the loss of residents' utility in enjoying education at the micro level. At the same time, there are urban problems such as traffic congestion and high housing prices in areas with concentrated educational resources; On the macro level, it will reduce urban efficiency and hinder the realization of educational equity. The rural basic education resources in Wuhan City shows the spatial distribution characteristics of more in the north than in the south, and more in the west than in the east. The distribution of different types of basic education facilities is imbalance, especially the number of middle schools is less. It shows that we should pay attention to the synchronous optimization of the structure of basic education resources in the allocation of education resources. Then, the areas with high spatial accessibility are mainly distributed in Chengguan Town and its surroundings, which have the characteristics of multi center layout, reflecting that the spatial distribution of basic education is closely related to the development status and level of the region. Population, economy, and education development levels are the main factors affecting the spatial distribution of rural basic education resources. The cause of the unbalanced distribution of compulsory education resources is not only the scale contradiction, but also the mobility of school-age population, urban development stage, and economic development level will affect the carrying capacity of compulsory education, which lead to the unbalanced spatial distribution of compulsory education resources. It is necessary to comprehensively consider the resource quantity, resource allocation efficiency and resource spatial balance, and establish a spatial optimization mechanism related to degree and population mobility.
Based on the above conclusions, the following suggestions are put forward: ①Establishing a flexible financial management system for compulsory education to fully release the dividends of supply-side reform. The study found that the basic education re-sources in the north are higher than those in the south and the west is higher than those in the east, and the distribution of all types of basic education facilities is not balanced. This is mainly due to the differences in social economy and population size. With the continuous progress of new urbanization in Wuhan, the trend of population flow to urban areas is further strengthened, and the mechanical migration of population will have a more significant impact on the regional population size and structure. In this case, the original "two main" compulsory education management system exposed the lack of flexibility, and it is difficult to properly solve the problem of migrant children enrollment limitations. Therefore, we should adhere to the people-oriented new urbanization and establish a flexible financial management system for compulsory education. ②Supporting educational facilities flexibly according to the population structure and demand difference of different regions. On the one hand, when selecting and constructing basic educational facilities, attention should be paid to reducing the overlap of service radius, improving the efficiency of resource utilization and promoting the accessibility of educational resources. At the same time, the scale of educational facilities should gradually transition from the "urban-rural" two levels to the more reasonable "urban-town-village" three levels, promoting the coordinated development of educational resources. ③The government should increase the proportion of high-quality teachers continuously in rural areas, incline policies such as talent introduction and capital investment to rural areas, focus on teacher training; Then, we should improve school hardware facilities to promote the balanced development of rural schools, build an efficient communication platform between urban schools and rural schools, promote the interconnection of high-quality education re-sources, and ensure the all-round quality balance of compulsory education opportunities, processes, quality and results.
Point 3: The references in the introduction should be in brackets. The quality of Figure 4 should be improved and possibly put it in color like Figure 3.
Response 3: Thanks for your comment. The references in the introduction have been checked and are all in brackets. The quality of Figure 4 has modified. Please see the attachment.

Round 2
Reviewer 2 Report
No suggestions to add, authors have improved the manuscript, especially in discussing limitations and paving the ground for future research.